# Unraveling Mechanisms and Impact of Microbial Recruitment on Oilseed Rape (*Brassica napus* L.) and the Rhizosphere Mediated by Plant Growth-Promoting Rhizobacteria

**DOI:** 10.3390/microorganisms9010161

**Published:** 2021-01-12

**Authors:** Ying Liu, Jie Gao, Zhihui Bai, Shanghua Wu, Xianglong Li, Na Wang, Xiongfeng Du, Haonan Fan, Guoqiang Zhuang, Tsing Bohu, Xuliang Zhuang

**Affiliations:** 1School of Life Sciences, University of Science and Technology of China, Hefei 230026, China; lypoppy@mail.ustc.edu.cn; 2CAS Key Laboratory of Environmental Biotechnology, Research Center for Eco-Environmental Sciences, Chinese Academy of Sciences, Beijing 100085, China; jiegao@rcees.ac.cn (J.G.); zhbai@rcees.ac.cn (Z.B.); shwu@rcees.ac.cn (S.W.); xlli_st@rcees.ac.cn (X.L.); zang09220601@163.com (N.W.); xfdu_st@rcees.ac.cn (X.D.); fanhnan@foxmail.com (H.F.); gqzhuang@rcees.ac.cn (G.Z.); 3College of Resources and Environment, University of Chinese Academy of Sciences, Beijing 100049, China; 4CSIRO Mineral Resources, Kensington, WA 6151, Australia; Qing.Hu@csiro.au

**Keywords:** PGPR, rhizosphere, plant growth stage, dynamic rhizobacteria community, network analysis

## Abstract

Plant growth-promoting rhizobacteria (PGPR) are noticeably applied to enhance plant nutrient acquisition and improve plant growth and health. However, limited information is available on the compositional dynamics of rhizobacteria communities with PGPR inoculation. In this study, we investigated the effects of three PGPR strains, *Stenotrophomonas rhizophila*, *Rhodobacter sphaeroides*, and *Bacillus amyloliquefaciens* on the ecophysiological properties of Oilseed rape (*Brassica napus* L.), rhizosphere, and bulk soil; moreover, we assessed rhizobacterial community composition using high-throughput Illumina sequencing of 16S rRNA genes. Inoculation with *S. rhizophila*, *R. sphaeroides*, and *B. amyloliquefaciens*, significantly increased the plant total N (TN) (*p* < 0.01) content. *R. sphaeroides* and *B. amyloliquefaciens* selectively enhanced the growth of *Pseudomonadacea* and *Flavobacteriaceae*, whereas *S. rhizophila* could recruit diazotrophic rhizobacteria, members of *Cyanobacteria* and *Actinobacteria*, whose abundance was positively correlated with inoculation, and improved the transformation of organic nitrogen into inorganic nitrogen through the promotion of ammonification. Initial colonization by PGPR in the rhizosphere affected the rhizobacterial community composition throughout the plant life cycle. Network analysis indicated that PGPR had species-dependent effects on niche competition in the rhizosphere. These results provide a better understanding of PGPR-plant-rhizobacteria interactions, which is necessary to develop the application of PGPR.

## 1. Introduction

The rhizosphere is one of the most complex Critical Zone (CZ) ecosystems in which plant growth-promoting rhizobacteria (PGPR) plays an indispensable role [1], as these organisms can promote plant growth or health, either directly or indirectly [2], through a wide variety of mechanisms including biological nitrogen fixation [3], phosphate solubilization [4], siderophore production [5], production of 1-aminocyclopropane-1-carboxylate deaminase (ACC) [6], phytohormone production [7], quorum sensing (QS) [1], induction of systemic resistance [8], and interference with pathogen toxin production [9]. PGPR inoculated into the rhizosphere could interact with the beneficial bacteria already present and shape the bacterial community composition, promote beneficial plant-microbe symbioses, and enhance plant growth [10] and plant pathogen defense [11].

The use of soil PGPR and their associated beneficial effects has the potential to improve plant growth and quality characteristics. *Bacillus amyloliquefaciens* is widely found in soils and is a common PGPR [12]. *B. amyloliquefaciens* can suppress *Fusarium* disease by manipulating rhizosphere microbial community composition and stimulating potentially beneficial taxa [13]. It has also been reported that treatment with *B. amyloliquefaciens* could markedly improve the plant growth and resist plant viral disease by changing rhizosphere microbial community structure and enhancing plant systemic resistance [14]. *B. amyloliquefaciens* has been shown to be capable of pathogen suppression including root colonization by the wilt pathogen *Ralstonia solanacearum* QL-RFP [12], anthracnose disease in chili [15], and *Pectobacterium carotovorum* subsp. *carotovorum* (*Pcc*), the cause of soft rot in Chinese cabbage [16]. Wu [10] found that *B. amyloliquefaciens* had the dual effects of promoting plant growth and reducing N_2_O emissions by changing the rhizosphere microbial community structure. These studies indicate the complex interactions that exist between PGPR strains and the native rhizosphere microbial community.

Phototrophic microorganisms are one type of PGPR, which can promote plant growth [17]. *Rhodobacter sphaeroides* represents one of the best-studied photosynthetic organisms [18] which can produce carotenoids [19], coenzyme Q10 [20], and superoxide dismutase [21]. *R. sphaeroides* is widely distributed in a variety of soil environments [22] and, within the rhizosphere, it is capable of soil restoration [23]. *R. sphaeroides* can increase ascorbic acid content in tomato fruit, promote root growth, increase leaf number, chlorophyll, carotenoid content, and average crop weight [24,25]. Moreover, Kensuke Kondo [26] found that *R. sphaeroides* application in sterile soil showed different effects on spinach growth, compared to its application in unsterilized soil, therefore the effect of *R. sphaeroides* application may due to interactions with other soil microorganisms.

Moreover, *Stenotrophomonas rhizophila* is another model of PGPR, especially under salt stress conditions [27,28,29]. *S. rhizophila* was first isolated from the rhizosphere of oilseed rape, and its detailed biochemical properties were described by Wolf et al. [30]. It can promote sweet pepper and tomato growth in gnotobiotic systems [31] and isolates of this species are known to produce volatile antifungal compounds [32]. Egamberdieva, et al. [33] found *S. rhizophila* and Bradyrhizobium built beneficial associations in the rhizosphere, acting synergistically to promote plant growth and nutrient uptake. Schmidt [31] used single-strand conformation polymorphism (SSCP) of 16S rRNA and ITS genes to assess the potential of *S. rhizophila* to inhibit plant pathogens. However, the detection of SSCP was limited, and the dynamic structural changes in the rhizosphere community after the addition of *S. rhizophila* were not revealed.

Previous studies on the PGPR mentioned above describe how they can manipulate the composition of the rhizosphere bacteria community. Taxa affected by PGPR have been found to be beneficial for plant growth and resistance to biotic and abiotic stresses. Understanding the interaction among different microbial species in the rhizosphere and their responses to PGPR is essential.

For the present study, we choose *B. amyloliquefaciens, R. sphaeroides,* and *S. rhizophila* as the three experimental PGPR strains. We collected rhizosphere soil from oilseed rape plants under three different PGPR treatments and at four growth stages. By determining the properties of plants and soil and monitoring the rhizosphere microbial community, we aimed to address the following issues: (i) the growth-promoting effect(s) of different microbial species; and (ii) how the assembly dynamics of plant rhizobacteria communities are affected by the inoculated PGPR. This study may improve the understanding of rhizobacteria in sustainable plant-growth promotion technologies, and the role of PGPR in biogeochemical processes of the Critical Zone.

## 2. Materials and Methods

### 2.1. PGPR Inoculants and Plants

*Stenotrophomonas rhizophila* DSM 14405^T^ (stored in ATCC)*, Rhodobacter sphaeroides* EBL0706 (stored in CGMCC)*, and Bacillus amyloliquefaciens* FZB42 (stored in CGMCC) cultures were grown aerobically at 30 °C and 150 rpm in LB Media for 20 h, 24 h, and 29 h respectively, until their density reached 2.0~3.0 × 10^8^ CFU mL^−1^. The bacterial cultures were pelleted by centrifugation and washed twice with sterile water and suspended in sterile water to prepare PGPR inocula respectively.

The cultivar of oilseed rape used in this study was Jingguan (*Brassica campestris* L. cv. Jingguan). Seeds were soaked in 2% NaClO for 10 min, then thoroughly rinsed with sterile water three times, and stored at 4 °C for vernalization in the dark. After 48 h, seeds were placed into an illuminated incubator (20 °C, 12 h/12 h day/night) for seven days.

### 2.2. Greenhouse Pot Experiment

Four treatments were set up: Sr (*S. rhizophila*), Rs (*R. sphaeroides*), Ba (*B. amyloliquefaciens*), and CK (control check) treatment located at Gudaoxifeng organic farm, in Beijing, China (N 40°4′25.10″, E 116°12′38.41″), each treatment was replicated three times, twelve pots (41 cm length, 27 cm width, 16 cm height) were filled with the homogenized soil in the greenhouse. The amount of soil per pot was approximately 14,169.6 g (Appendix A). The three PGPR treatments Sr, Ba, and Rs received each 1 × 10^7^ cfu/g of their respective organisms, the CK treatment only received sterile water. After bacteria were well mixed with the soil, plant seedlings were transferred to the pots, forty plants per pot. The plant growth process took 59 days, at 4 °C~30 °C, from 17 November 2018 to 14 January 2019.

### 2.3. Plant Physicochemical Parameters

On day 59 of plant growth, ten individual plants per pot in each treatment were collected and placed into sterile zip-lock bags and transported on ice to the laboratory. The content of chlorophyll, soluble starch, soluble protein, and Vitamin C was determined in the youngest fully opened leaf of the plants. Chlorophyll was quantified by SPAD-502 plus meter (Minolta Camera Co., Osaka, Japan) [34]. Soluble starch concentrations were determined using the Anthrone assay [35]. Soluble protein was extracted and quantified using the Bradford assay [36]. Vitamin C was determined using the FRASC assay [37]. The remaining above-ground biomass was harvested, oven-dried at 60 °C for two days, and weighed [34]. The total C (TC), total N (TN), and total S (TS) of plants were measured using an elemental analyzer (Vario EL III, Elementair, Hanau, German) [10].

### 2.4. Rhizosphere and Bulk Soil Collection and Physicochemical Parameters

At day 5 (second leaf), day 17 (fourth leaf), day 23 (sixth leaf), and day 59 (tenth leaf) during plant growth, five bulk soil cores were taken at random from the topsoil layer (1–10 cm) and pooled as one bulk soil sample, then transferred into sterile 50 mL tubes. Five individual plants per pot in each treatment were dug out with their surrounding soil and transferred into sterile zip-lock bags. Tubes and bags were transported on ice to the laboratory. Bulk soil was shaken off the roots [38], which were then cut from the plant and thoroughly washed three times with sterile PBS buffer by extensive shaking. These washes were filtered through 0.22 µm pore size membranes and substances remaining on the membrane after filtration were considered as the rhizosphere soil. Membranes and attached particles were transferred to 2 mL tubes (Appendix A). Fresh bulk and half of the rhizosphere soil were analyzed for their physicochemical properties; the remaining rhizosphere soil was flash-frozen in liquid nitrogen and stored at −40 °C until DNA extraction.

The physical and chemical properties of the day 59 bulk and rhizosphere soil samples were determined in sextuplicate per treatment, all soil samples were passed through a 2-mm mesh. Soil pH was determined in samples with soil to water ratio of 1:2.5 (*w*/*v*) using a pH-meter (LE438, METTLER TOLEDO, Greifensee, Switzerland). The nitrate (NO_3_^−^) and ammonium (NH_4_^+^) content of the soil was extracted by 2 M KCl with soil to solution ratio of 1:10 (*w*/*v*) and measured using a Continuous Flow Analyzer (SAN++, Skalar, Breda, Holland). The TC, TN, and TS of soil were measured using an elemental analyzer (Vario EL III, Elementair, Hanau, German). Content of Ca, Mg, Fe, Mn, Al, K, and P in soil were determined by inductively coupled plasma spectrometer (ICPE-9820, Shimadzu, Kyotao, Japan).

### 2.5. Evaluation of Plant Growth Promotion Characteristics

Analyses of the plant growth-promoting capabilities of the three strains in this study focused on nitrogen fixation, phosphorus solubilization, indoleacetic acid (IAA) detection, siderophore production, ACC deaminase activity, and biofilm synthesis. Burks Medium was used to test the nitrogen fixation and Pikovskaya’s Broth medium was used to determine the phosphate-solubilizing capacity [39]. The colorimetric Salkowski assay was used to measure IAA [40] production. Blue agar chrome azurol S (CAS) assay [41] detected the production of siderophores. ACC deaminase activity was determined as reported by Hansen and Moller [35]. Biofilm production was determined by crystal violet staining [42].

### 2.6. DNA Extraction and 16S rRNA Sequencing

The DNA of rhizosphere soil was extracted using the DNeasy PowerSoil Kit (QIAGEN, Hilden, Germany), according to the manufacturer’s protocol. Six parallel samples were extracted from each treatment at four time points (day 5, 17, 23, and 59). DNA quality and concentration were measured on Nanodrop 2000 (Thermo Scientific, Waltham, MA, USA).

V3-V4 region of the bacterial 16S rRNA gene was amplified using the primer pair 338F (5′-ACTCCTACGGGAGGCAGCAG-3′) and 806R (5′-GGACTACHVGGGTWTCTAAT-3′). Each sample was amplified in a 20 μL reaction volume containing 2.5 units FastPfu Polymerase, 4 μL 5× FastPfu buffer, 2.5 mM dNTPs, 0.2 μL BSA (all TransGen, Beijing, China), 0.5 µM forward primer and reverse primer. PCR was performed using the following PCR program: 95 °C for 3 min, 27 cycles of 95 °C for 30 s, 55 °C for 30 s, 72 °C for 45 s, followed by 72 °C for 10 min. PCR quality was assessed by visualizing the amplicon on a 2% TAE agarose gel. Sequencing was carried out utilizing the Illumina MiSeq platform (Illumina, Inc., San Diego, CA, USA) at Shanghai Majorbio Biotechnology Co. Ltd (China).

Paired-end reads were assigned to the samples based on their unique barcodes, after which the barcode and primer were trimmed. Paired-end reads were merged using FLASH (V1.2.11, https://ccb.jhu.edu/software/FLASH/index.shtml) [43]. Quality filtering was performed to obtain high-quality clean sequences [44] by QIIME (V1.9.1, http://qiime.org/install/index.html) [45]. Chimera sequences were detected [46] using UCHIME (http://www.drive5.com/usearch/) [47]. Subsequently, all the retained sequences were analyzed by UPARSE (V7.0.1090, http://www.drive5.com/uparse/) [47]. The operational taxonomic units (OTUs) were created based on a 97% similarity. The RDP Classifier (V2.11, https://sourceforge.net/projects/rdp-classifier/) [48] and Silva Database (https://www.arb-silva.de/) [49] were used to assign taxonomy of representative sequences. To account for differences in sequencing depth, all samples were resampled to the lowest number of sequences (37,871) among all samples. Alpha and beta diversity were calculated using Mothur (V1.30.2, https://www.mothur.org/wiki/Download_mothur) and QIIME (V1.9.1) respectively. R (V 2.15.3) software was used. All the raw sequences in this study were deposited in the NCBI Sequence Read Archive, with the accession number PRJNA622875.

### 2.7. The Quantification PCR of the PGPR and N Cycling Functional Genes in the Rhizosphere 

To detect PGPR colonization and quantify functional genes *ureC*, *nifH, and nxrA* in the rhizosphere, six plasmid standards containing target regions for each inoculants strain were constructed for qPCR which was done following the previously described protocols [50,51,52,53,54]. The specific gene sequences were amplified from extracted DNA with primers listed in Table 1. In the case of *S. rhizophila*, a sequence unique to this organism [55] was selected for the quantification target gene. The amplified products were run on a 1% agarose gel to confirm the specificity of the amplification, and then the gel was cut and fragment purified by the TIANgel Midi Purification Kit following the manufacturer’s protocol (TIANGEN, Beijing, China). The purified PCR products were sequenced and cloned into the pGEM-T Easy Vector (Promega, Madison, WI, USA), and then transformed into the *Escherichia coli* DH5α competent cells (TIANGEN, Beijing, China). The plasmids were then extracted by the TIANprep Mini Plasmid Kit (TIANGEN, Beijing, China) and concentrations determined by a Nanodrop 2000 (Thermo Fisher, Waltham, MA, USA) system. The plasmids used as standards for quantitative analysis were extracted from the correct clones of target genes, with confirmation by sequencing (Ruibio, Beijing, China). Copy numbers of target genes were calculated directly from the concentrations of extracted plasmid DNA. Standard curves were generated using triplicate 10^−^ fold serial dilutions of known copy numbers of plasmid DNA.

The *S. rhizophila* qPCR reaction mixture contained 12.5 μL 2X TB GreenPremix Ex Taq (TaKaRa, Shiga, Japan), 0.2 μM forward and reverse primer, 2 μL template DNA, adjusted to a total 25 uL with sterile water. Thermal cycling consisted of an initial denaturation at 95 °C for 30 s, followed by 40 cycles of 95 °C for 5 s, 58 °C for 10 s, and 72 °C for 10 s. All qPCR was conducted by CFX96 Real-Time PCR Detection System (Bio-Rad, Hercules, CA, USA).

### 2.8. Network Construction with RMT-Based Approach and Topological Analysis

To elucidate rhizobacterial interactions in the presence of PGPR and assess changes in bacterial community assembly, molecular ecological networks were constructed based on the random matrix theory (RMT) method [56] in the molecular ecological network analysis pipeline (MENA, http://ieg4.rccc.ou.edu/mena/main.cgi) following the process as described in a previous study [56]. Bacterial OTUs of six samples per treatment were kept without log-transformation prior to carrying out the Spearman rank correlation matrix (r value). The networks were visualized using Cytoscape (V3.7.2).

### 2.9. Data Analysis

Data obtained from each treatment were statistically analyzed by one–way analysis of variance (ANOVA) and the means were compared using the post–hoc Tukey’s HSD test for multiple comparisons test for mean separation. Differences at *p* < 0.05 were considered as significant. The analyses were performed using the IBM SPSS 25 software.

## 3. Results

### 3.1. PGPR Promote Plant Growth and Enhance the Nutrient Availability in the Rhizosphere

All three PGPR inoculants significantly increased TN content in the plant (Table 2). The *Stenotrophomonas rhizophila* and *Bacillus amyloliquefaciens* treatments significantly increased the soluble protein content in the plant, whereas the *Rhodobacter sphaeroides* treatment caused a significant decrease of TC compared to the CK treatment.

Sr treatment significantly increased the concentrations of Al (5.45%), K (10.00%), P (24.37%), NH_4_^+^ (473.70%), and NO_3_^−^ (62.77%) in rhizosphere soil, as compared to the bulk soil (Table 3). Rs treatment significantly increased the concentrations of K (7.08%), NH_4_^+^ (250.95%), and TC (28.17%) in rhizosphere soil compared with bulk soil (Table 3). Ba treatment significantly increased NH_4_^+^ (142.64%) content but decreased the concentrations of Ca (14.06%), Mg (34.20%), Fe (14.11%), Mn (12.08%), Al (4.72%), and NO_3_^−^ (38.21%). The content of TS in rhizosphere soil significantly decreased compared with bulk soil in all PGPR treatments. There was no statistical difference between rhizosphere and bulk soil of the CK treatment, except for the increase of NH_4_^+^ (476.96%) content. 

Therefore, the addition of PGPR enhanced the differences in physical and chemical properties between rhizosphere and bulk soil, increasing the nutrient availability of rhizosphere soil.

Comparison of rhizosphere chemical properties of PGPR and CK treatment showed that the addition of *S. rhizophila* and *R. sphaeroides* significantly increased P concentration, 24.05% and 38.64% greater than CK treatment respectively. While the Sr treatment significantly decreased the Ca (7.47%) and Mg (16.18%) content and Rs treatment significantly decreased NH_4_^+^ (49.79%) and NO_3_^−^ (26.02%) content. Ba treatment significantly decreased the concentrations of Ca (185.38%), Mg (168.55%), Fe (189.84%), Mn (187.74%), NH_4_^+^ (143.47%), and NO_3_^−^ (157.93%). Thus, the application of PGPR changed the rhizosphere soil physicochemical characteristics.

### 3.2. Identification of PGPR Promoting Abilities

Results of the growth-promoting capabilities of the three PGPR strains are shown in Appendix A. *B. amyloliquefaciens* showed ACC deaminase activity and the ability to fix nitrogen. *B. amyloliquefaciens* and *S. rhizophila* were both able to dissolve phosphorus and produce biofilms. Siderophore production was observed in *S. rhizophila* and *R. sphaeroides*.

### 3.3. Low, Persistent Colonization of PGPR Strains in the Rhizosphere

In order to trace the colonization of the three PGPR inoculants in the rhizosphere, quantitative PCR was implemented. *R. sphaeroides* and *B. amyloliquefaciens* multiplied and accumulated in the rhizosphere soil of plants during the initial stage of plant growth, and reach their abundance peak at approximately ten days (Figure 1a), after which their abundance gradually decreased. On day 59, the copy number per gram of rhizosphere soil for these organisms was 9.13 × 10^6^ and 3.76 × 10^5^, respectively. The abundance of *S. rhizophila* in rhizosphere soil fluctuated strongly and displayed a decreasing tendency during the initial stage of plant growth. The copy number in the rhizosphere soil was 4.9 × 10^6^ g^−1^ on day 5, which was approximately half of the initial Sr dosage, the highest abundance was 7.3 × 10^6^ g^−1^ during the interim stage, then the abundance reached a relatively stable plateau period with an abundance of 3.4 × 10^6^ g^−1^ on day 23. On day 59, the copy number of *S. rhizophila* in the rhizosphere soil was 3.0 × 10^6^ g^−1^. In brief, the abundances of the three PGPR inoculants did not remain stable during the five-week period after inoculation. qPCR results indicated that *B. amyloliquefaciens* and *R. sphaeroides* had stronger rhizosphere colonization abilities than *S. rhizophila*, but none of the strains could sustain colonization of the rhizosphere.

### 3.4. S. rhizophila Increased the Ammonification in the Rhizosphere

To understand the effects of PGPR inoculants on N cycling of rhizosphere microbial communities, functional genes associated with N cycling, ammonification (*ureC*), N_2_ fixation (*nifH*), and nitrification (*nxrA*) [57] were quantified. The abundance of *ureC* gene showed a gentle upward trend with the growth of the plant in Rs, Ba, and CK treatments (Figure 2 and Appendix A). Compared with CK, the abundance of *ureC* began to increase significantly at the beginning of inoculation, the *ureC* gene increased 136.72% (*p* < 0.01), 281.33% (*p* < 0.001), and 173.62% times (*p* < 0.001) in Sr treatment on days 5, 17, and 23, respectively (Appendix A). This indicated that the application of *S. rhizophila* increased the ammonification in rhizosphere soil. The abundance of the *nifH* gene (Appendix A) showed an overall upward trend in all treatments and on days 5, 17, and 23 the *nifH* gene abundance was the highest in Sr treatment, with 9.62 × 10^11^ g^−1^, 1.58 × 10^12^ g^−1^, and 2.28 × 10^12^ g^−1^ copies respectively, but there was no significant difference among them. The abundance of the *nxrA* gene showed a slowly increasing trend with plant growth and there was no significant difference between the treatments.

### 3.5. Rhizobacterial Community Assembly Is Driven by Plant Growth Stage and PGPR

Illumina sequencing generated a total of 5,277,380 sequences that could be classified into 11147 OTUs belonging to 47 phyla and 121 classes after the removal of ambiguous, short, and low-quality reads and singleton OTUs. Microbial richness and diversity were estimated by the Chao and Shannon indices (Figure 1b). Species richness indices indicated an unsteady reduction in the number of rhizosphere OTUs during plant growth. In all samples analyzed, the species richness was significantly reduced on day 59 as compared with day 5 (Student’s *t*-test). Similarly, bacterial community diversity was reduced during plant growth except in the Sr treatment (Student’s *t*-test). In particular, the student’s *t*-test for Chao and Shannon indices showed that on day 59 (Appendix A), the OTUs richness and diversity of the Sr treatment were significantly higher than those in the other three treatments.

The majority of the bacterial sequences observed in the rhizospheres of the four treatments during plant growth belonged to the phyla *Proteobacteria* (23.66%~53.86%), *Bacteroidetes* (8.74%~31.68%), *Actinobacteria* (8.22%~16.85%), *Acidobacteria* (5.12%~9.64%), *Chloroflexi* (4.45%~40.8%), *Firmicutes* (1.53%~6.16%), *Gemmatimonadetes* (0.95%~2.84%), and *Cyanobacteria* (0.32%~18.67%). *Proteobacteria* and *Bacteroidetes* largely dominated the rhizosphere bacterial community dynamics (Figure 3a). During CK treatment plant growth, the rhizosphere was significantly enriched in *Bacteroidetes*, while the relative abundances of *Chloroflexi*, *Firmicutes*, *Acidobacteria*, and *Gemmatimonadetes* decreased (Appendix A). The addition of *B. amyloliquefaciens* significantly increased the relative abundance of *Proteobacteria*, while the addition of *R. sphaeroides* significantly increased the proportion of *Proteobacteria* and *Bacteroidetes*. The inoculation of *S. rhizophila* primarily increased the relative abundance of *Actinobacteria* and *Cyanobacteria* during plant growth (Appendix A). Wilcoxon rank-sum test also revealed that compared to the CK treatment, the relative abundances of *Proteobacteria* were significantly increased in Ba and Rs treatments during the final stage of plant growth (day 59). Sr treatment significantly increased the relative abundance of *Acidobacteria*, *Gemmatimonadetes*, and *Cyanobacteria* while decreasing the relative abundance of *Bacteroidetes* on day 59 (Appendix A). At the family level, the relative abundances of *Flavobacteriaceae* (Figure 3c) and *Pseudomonadacea* (Figure 3b) were significantly higher in the Rs treatment compared to CK treatment, meanwhile, the relative abundances of *Pseudomonadacea* and *Rhizobiaceae* (Figure 3d) were significantly higher in Ba treatment compared to CK treatment on day 59. The inoculation of *S. rhizophila* reduced the abundances of *Flavobacteriaceae*, *Pseudomonadacea*, and *Rhizobiaceae* compared to CK treatment, however, *S. rhizophila* augmented many *Chloroflexi* species (Appendix A).

Non-metric multidimensional scaling (NMDS) analysis showed that the shifts in rhizosphere soil bacterial community composition increased with time (Figure 4a); and that the bacterial community was obviously different in each of the four treatments on day 59. Most significantly, NMDS analysis showed a clear separation between Sr treatment and the other three treatments on day 59. These results showed that samples from different treatments were separated from each other and that community structures were altered during plant growth.

Bipartite association networks were used to visualize the associations between OTUs in the four treatments (Figure 4b). The quantity of OTUs specific to each treatment changed significantly over the four stages. On day 5, the number of OTUs unique to each treatment group was relatively balanced, however by day 59, OTU numbers unique to each treatment group changed dramatically. Among the top 200 species in abundance, at OTU taxonomic level, the number of unique OTUs in the Sr treatment increased from 30 (day 5) to 133 (day 59), a large proportion of which belonged to *Proteobacteria* (40), *Chloroflexi* (29), *Actinobacteria* (22), *Bacteroidetes* (16), and *Acidobacteria* (15). The number of OTUs unique to the Ba, Rs, and CK treatments had decreased by the final stage (day 59) compared with the initial number (day 5). In particular, the Rs treatment had only three unique OTUs during the final stage.

### 3.6. Dynamic Rhizobacteria Interactions during Plant Growth

Phylogenetic molecular ecological networks (pMEN) were constructed to identify the interactions within bacterial communities during plant growth with PGPR inoculants. In regard to network topological indices, higher average connectivity (avgK) indicates a more complex network [58], average clustering coefficient (avgCC) measures the extent of module structure present in a network [59], and a smaller average geodesic distance (GD) means all the nodes in the network are closer. A modularity index value of >0.4 suggested that the networks had a typical module structure [60]. Thus, rhizosphere communities underwent dynamic changes during plant growth, with the Sr treatment network soil showing more complexity than the other treatments on days 5 and 23; the CK treatment was more tightly linked than the PGPR treatments on day 17; and rhizosphere species were linked in a more complex manner in the Ba treatment on day 59. All networks showed high modularity, which is beneficial for increasing the stability of interaction networks and helping microbial communities resist environmental changes [61] (Appendix A). Total node number, average connectivity (avgK), average clustering coefficient (avgCC), and average geodesic distance (GD) indicated species interactions involved in the three PGPR treatments changed in a dissimilar manner. The pMENs were used to visualize the interactions among nodes at the genus level (Figure 5). The relative abundance of *Chloroflexi* in the networks was relatively high in the early stages and decreased over time. On day 17, *Cyanobacteria* appeared in the networks. In the final stage, *Pseudomonas* and *Flavobacterium* accounted for a greater proportion of links than *Chloroflexi* and were responsible for the two of the largest proportions of links in the Rs (7.83%, 9.22%), Ba (6.61%, 6.61%), and CK (6.79%, 6.73%) networks on day 59 respectively. In contrast, the Sr treatment network had lower node proportions of *Pseudomonas* and *Flavobacterium* on day 59 as compared with the other three networks. *Pseudomonas* and *Flavobacterium* showed positive interactions in all treatments.

To further distinguish keystone species in the interaction networks, all nodes were divided into four categories (peripherals, connectors, module hubs, and network hubs) according to the Zi (within-module connectivity) and Pi values (among-module connectivity) in ZP-plot (Appendix A). The majority of nodes could be classified into peripherals with most of their links within their modules, accounting for 98.50~100% of nodes. Module hubs represent nodes highly connected within their own modules that could be regarded as central species within each unit. The sixty-four nodes identified as module hubs and were mainly from *Proteobacteria*, *Chloroflexi*, *Actinobacteria*, *Acidobacteria*, *Bacteroidetes,* and *Gemmatimonadetes*. Connectors represented nodes that were highly linked to other modules acting as bridges. The twenty-two nodes identified as connectors mainly included *Proteobacteria*, *Chloroflexi*, *Actinobacteria*, and *Firmicutes* (Appendix A).

## 4. Discussion

The beneficial effects of PGPR on plant growth are well documented [62,63]. However, the role of PGPR is not solely implemented through the direct effect of a single bacterial strain, but by the molecular dialogue established among multiple soil microorganisms and the plant [64]. Our research was based on a fifty-nine-day greenhouse experiment in which agricultural management schemes (e.g., watering regime and climatic conditions) were the same for all treatments (Sr, Rs, Ba, and CK). Fifty-nine days was considered a suitable time span [10] to ascertain the combined effects of PGPR on plant performance. We assessed the physicochemical parameters of both plant soil, conducted PGPR colonization tests, and quantified N cycling genes, and utilized the information of 16S rRNA genes to analyze the rhizobacteria diversity, composition, and molecular ecological networks.

### 4.1. Plant Growth Stages Determine Rhizobacterial Community Composition

In general, except for the Sr treatment, the abundance of *Proteobacteria* and *Bacteroidetes* were significantly enriched in the rhizosphere soil compared to the early samples (day 5), while, in contrast, the relative abundances of *Chloroflexi*, *Acidobacteria*, *Firmicutes*, and *Gemmatimonadetes* in the rhizosphere were reduced compared to the initial stage (Figure 3a). The dynamic changes in bacterial abundances of CK treatment altered gradually over time. This proved that plant growth established the rhizosphere community [65]. An increased abundance of *Proteobacteria* and *Bacteroidetes* has been described in previous studies [66,67]. Members of the *Proteobacteria* are rhizosphere colonizers and fast-growing r-strategists, which respond positively to the rhizosphere [68]. *Bacteroidetes* are also fast-growing, capable of quick organic matter decomposition, and often increase in abundance after planting [69]. On the contrary, *Chloroflexi*, *Firmicutes, Acidobacteria*, and *Gemmatimonadetes* decreased in relative abundance in the rhizosphere as the plants grew (Appendix A).

The decrease in the relative abundances of *Chloroflexi*, *Actinobacteria*, and *Firmicutes* in the rhizosphere could be due to competition with fast-growing *Proteobacteria* and *Bacteroidetes*, or other microbes, for resources or from microbe-microbe inhibition.

Previous research has confirmed that plants are able to control the composition of their rhizosphere microbiome [70] to select for specific microbial functions [71] and to support, restrict, or terminate microbial growth and activity [1]. Meanwhile, bacteria adapt to the rhizosphere by making strategic adjustments through motility, chemotaxis, quorum sensing [1], lipopolysaccharide synthesis [72], increased biofilm formation, or altering substrate utilization profiles [73], reshaping the bacterial communities [1]. Hiltner [74] proposed that rhizosphere microbial communities impact plant nutrition and health [75] and that, conversely, these communities may be affected by the stage of plant growth [76]. Based on the analysis of CK treatment plant rhizosphere communities during the four stages, our results provide support for previous studies, where the stage of plant growth was determined to be an important determinant of rhizosphere microbial composition [76,77].

### 4.2. R. sphaeroides and B. amyloliquefaciens Increase the Selective Enrichment of Beneficial Bacteria in the Plant Rhizosphere

In regard to Rs and Ba treatments, the enrichment of *Proteobacteria* and *Bacteroidetes* in the rhizosphere was strengthened by the application of *R. sphaeroides* and *B. amyloliquefaciens.* Rs treatment significantly increased the relative abundance of *Pseudomonadaceae* (*Proteobacteria*) and *Flavobacteriaceae* (*Bacteroidetes*), and Ba treatment significantly increased the relative abundance of *Pseudomonadaceae* and *Rhizobiaceae* (classified as *Proteobacteria*). 

Network analysis further confirmed *Pseudomonas* and *Flavobacterium* played crucial roles in bacterial interactions during the final stage of Rs and Ba treatments (Figure 5). There is evidence that interactions among bacteria play an important role in community dynamics or assembly [78]. Additionally, the bacterial community assembly rules reflecting ecological interaction in the rhizosphere, such as cooperation, competition, and niche partitioning [79], were fully demonstrated by the networks, providing a reference for how PGPR affects the rhizosphere bacterial community. The networks presented dissimilar dynamic changes during the four stages, Rs and Ba treatments had a tighter and more complex network of rhizobacteria, whereas Sr treatment showed higher modularity during the final stage (Appendix A). The rhizosphere microbiome extends the capacity of plants to adapt to environmental changes, and the establishment of particular microbiome members in the rhizosphere can be regarded as niche colonization [1]. Specific conditions of organic soil, such as high organic matter content or the presence of anoxic habitats under waterlogged conditions [80], may provide additional niches for anaerobic bacteria [81] or facultative anaerobes (e.g., *Chloroflexi*). This could explain the high abundance of *Chloroflexi* throughout the experiment and its early occupation of an important ecological niche. Subsequently, *Chloroflexi* species were at a disadvantage during niche competition in Rs, Ba, and CK treatments, but had the dominant position in Sr treatment. *Cyanobacteria* were supplanted in niche competition by other species in Rs and Ba treatment. Moreover, positive connections existed within genera *Pseudomonas*, *Flavobacterium*, and *Rhizobium. Pseudomonas* are known to promote plant growth and enhance crop yield [82]; act as bio-control agents [83]; and recognize and send quorum sensing QS molecules [84], which play a fundamental role in shaping the rhizosphere microbial community as well as influencing plant development [1]. *Pseudomonas* and *Rhizobium* are among the most powerful phosphate solubilizers in soil [85]; and due to their nitrogen fixation capabilities, *Rhizobium* also maintains soil fertility, enhancing crop yields [86]. Additionally, *Flavobacterium* can produce plant hormone of the auxin class, IAA [87], and alter other rhizosphere enzyme activities [88]. 

We showed that the application of *R. sphaeroides* and *B. amyloliquefaciens* manipulated rhizobacterial communities to greatly enhance selective enrichment (e.g., *Pseudomonadaceae* and *Flavobacteriaceae*). Simultaneously, pMENs reveal the intense niche competition and succession of rhizobacteria in the Rs and Ba treatments. 

### 4.3. The Difference of S. rhizophila Application to the Rhizobacterial Community

Interestingly, the inoculation of *S. rhizophila* affected alpha-diversity and rhizobacterial microbiome dynamics differently. Reduced bacterial community richness and diversity has been reported for the rhizosphere of several other plant species [81,89], which was consistent with the results of Rs, Ba, and CK treatments (Figure 1b). It is believed that the decrease in OTU richness in the rhizosphere can be attributed to a homogenizing effect of rhizosphere processes that reduce niche dimensions [81]. Alternatively, a reduction in community diversity could result from altered species abundance distributions over time [90]. However, there was no statistical decrease in the diversity of Sr treatment. Moreover, the bipartite association network (Figure 4b) also demonstrated that the Sr treatment had higher bacterial community diversity during the final stage of plant growth. Overall, the application of *S. rhizophila* prevented the decline of rhizosphere species diversity over time.

*Proteobacteria* and *Chloroflexi* were core phyla in the Sr treatment rhizosphere microbiome throughout plant growth (Figure 3a). *S. rhizophila* remarkably increased the relative abundance of *Acidobacteria*, *Cyanobacteria*, and *Gemmatimonadetes* and reduced *Bacteroidetes* abundance as compared to CK treatment during the final stage (Appendix A). Sr treatment enriched *Actinobacteria* over time (Appendix A), and similar assembly change had previously been observed in the sanqi rhizosphere [91]. Moreover, *Chloroflexi* has been observed in the rhizosphere of many plants, including maize [89], soybean [92], and rice [77]. 

*Chloroflexi* are widely distributed in terrestrial and aquatic ecosystems [93] and many of their members take part in the nitrogen cycle either through nitrogen fixation [94] or nitrite oxidation [95]. *Cyanobacteria* are widely distributed in various environments and considered as plant growth promoters in maize [96] and wheat [97] rhizosphere systems. Like *Cholorflexi*, *Cyanobacteria* are able to fix nitrogen [98], they are also able to increase the availability of phosphorous and release auxins which promote plant growth [99], identifying them as a green option for sustainable agriculture [100]. Their ecological niche indicates a potentially important role in nitrogen fixation agroecosystems [101]. 

Furthermore, *Cyanobacteria* can produce H_2_ as a by-product of nitrogen fixation [102]. They also produce glycolate (a by-product of photorespiration under conditions of O2 supersaturation) during the daytime [103] and produce acetate and propionate during the nighttime (under anoxic conditions) [104]. There has been evidence that phototrophic *Chloroflexi* perform both photomixotrophy, when metabolic intermediates and electron donors, such as H_2_ are available during the daytime [102], mixotrophy, simultaneously incorporate both acetate and glycolate (Appendix A). Therefore, *Cyanobacteria* participate in the metabolic activities of *Chloroflexi* by mutualism. We speculate that *Chloroflexi* may be linked with the nutrient supplying capability of soil and plant growth promotion and that this group deserves more exploration in future studies.

*Acidobacteria*, believed to be K-strategists that would respond slowly to environmental perturbation [105], whose abundance is negatively correlated with nutrient availability [106], increased during the final stage of plant growth. We believe this increase in *Acidobacteria* abundance was due to the reduced soil organic matter and soil resources as plants grew during the final stage. Some members of *Actinobacteria* are also capable of nitrogen-fixation [107], biocontrol [108], enhancing nutrient availability, and regulating plant metabolism [109]. *Actinobacteria* may also be involved in the degradation of organic matter in the anoxic zones of paddy fields [110]. Additionally, some *Gemmatimonadetes*, which are abundant but poorly characterized bacteria in soil, have been identified to possess the capacity for nitrite to nitric oxide reduction [111].

Previously, Schmidt [31] showed that one of the *S. rhizophila* plant growth promotion strategies was to shape the fungal rhizosphere community. Our results suggest that another tactic is to reshape the rhizobacteria community by recruiting beneficial diazotrophic bacteria such as members of the *Cyanobacteria* and *Actinobacteria*.

### 4.4. PGPR Promote Plant Growth and Enhance Soil Nutrient Availability by Shaping the Rhizobacterial Community

Quantification PCR was performed to assess rhizosphere colonization and N cycling functional genes quantification, in the process, we established an accurate *S. rhizophila* species-specific qPCR system. The trends of the three PGPR’s colonizing abilities (Figure 1a) were in accordance with previous results [112], in that even if PGPR inoculants colonize during the initial plant growth stage, their persistence over time is not guaranteed. Notably, our study indicates that even if PGPR are present even at low levels during the final plant stage, they can still affect the rhizosphere community structure. Therefore, the addition of PGPR in the initial stage of plant growth could affect the rhizobacteria community over the course of a plant’s entire life.

Plant growth and productivity depend on the availability of nutrients at the soil-root interface [113]. The biogeochemical cycle of the entire N pool is very important to the level of soil fertility, and the N cycle is primarily regulated by microbial processes [114]. The role of PGPR inoculants with an impact on N utilization by plants has been widely described [115]. In our study, the abundance of ammonification gene *ureC* was increased in Sr treatment. This suggested that effective transformation of organic nitrogen into NH_4_^+^ by rhizobacteria, which was enriched by the inoculation of *S. rhizophila*, may contribute to higher N use efficiency in the plant. But ammonification may cause the loss of NH_3_ and N_2_O emissions [116], which deserves our attention. *R. sphaeroides* and *B. amyloliquefaciens* enhanced the selective enrichment of *Pseudomonadaceae* and *Flavobacteriaceae* in the rhizosphere soil to varying degrees; while *S. rhizophila* recruited diazotrophic *Cyanobacteria and Actionbateria* species increase in the amounts of TN in the plant significantly. Nitrogen is an imperative element for the proper growth and development of plants by playing a vital role in the biochemical and physiological functions of plants, including the generation of amino acids for proteins [117]. The addition of *S. rhizophila* and *B. amyloliquefaciens* significantly increased the content of soluble protein, which is an important indicator for detecting the plant enzyme activity and total metabolism [118]. Thus, we suggest that the three PGPR strains increased bioavailable N directly to the plant, and in turn could decrease the reliance on synthetic fertilizers for agriculture.

Moreover, the addition of PGPR resulted in pronounced changes in rhizosphere nutrient composition compared to bulk soil. The Sr treatment significantly increased the content of Al, K, P, and NO_3_^−^, while the Rs treatment observably increased the K and TC content in rhizosphere soil. However, the Ba treatment significantly decreased the Ca, Mg, Fe, Mn, Al, and NO_3_^−^ concentrations, possibly due to *B. amyloliquefaciens* enhanced nutrients absorption by plant root. Thus, different PGPR inoculants have different degrees of effectiveness in altering soil nutrients.

These results, derived from a model plant, could be used to improve sustainable productivity in agriculture but will require extensive research in other crops. Meanwhile, if there is a comparison of available element concentrations with bulk and rhizosphere soil, to show the absorption of nutrients directly, it will further our understanding of how PGPR plays a pivotal role in shaping the microbiome and conduct soil nutrients transformation.

## 5. Conclusions

The inoculation of PGPR increased the TN and soluble protein content in the plant and altered nutrient availability in the rhizosphere soil. Together, the stage of plant growth and PGPR drove rhizobacterial community composition. Based on quantitative PCR and the bacterial community structure analysis with four stages, it was demonstrated that, although the PGPR cannot colonize the rhizosphere in a persistent manner, the effect of PGPR on the rhizobacterial community was long-lasting. Our results reveal that different PGPR inoculants may lead to the different assemblages of rhizobacterial communities and rhizobacterial interactions during the process of plant growth: *R. sphaeroides* and *B. amyloliquefaciens* assist the plant to selectively enrich for beneficial bacteria to a greater degree by increasing the relative abundance of *Pseudomonadacea* and *Flavobacteriaceae*. *S. rhizophila* attracted diazotrophic members of *Cyanobacteria* and *Actinobacteria*, and stimulated ammonification. Simultaneously, we established a qPCR system for *S. rhizophila* and detected plant growth-promoting characteristics of *R. sphaeroides* for the first time. In summary, the three PGPR strains in this study promote plant growth and participate in biogeochemical processes by affecting the composition of the rhizobacterial community.

## 6. Patents

Application of *Stenotrophomonas rhizophila* in improving the rhizosphere soil and promoting plant growth. Application number 202010323710.3.

## Figures and Tables

**Figure 1 microorganisms-09-00161-f001:**
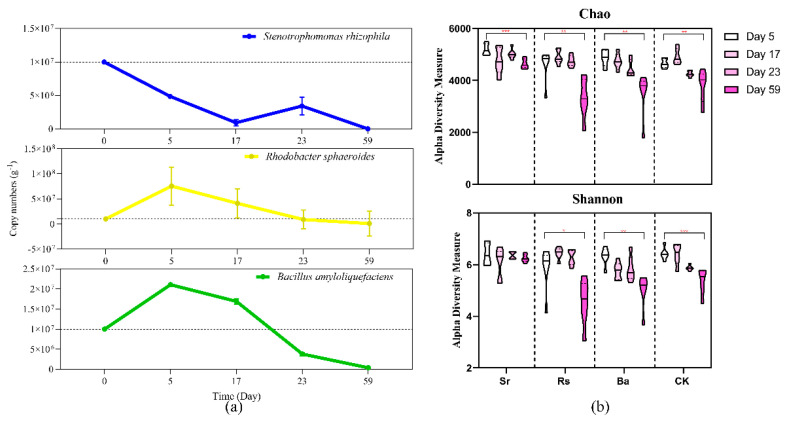
(**a**) Quantitative curve trends of PGPR strains from day 5 to day 59, the peripheral curve represents the error (SD) line; (**b**) Chao and Shannon index of alpha diversity for four treatments during four stages. Significant levels: ns; * *p* < 0.05; ** *p* < 0.01, *** *p* < 0.001.

**Figure 2 microorganisms-09-00161-f002:**
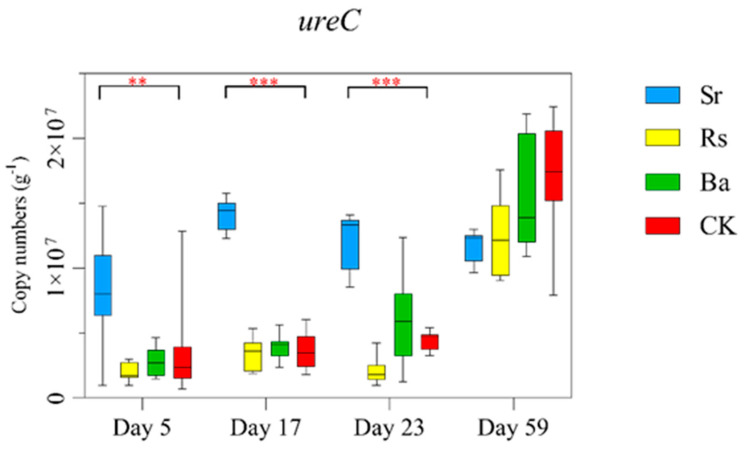
Differences in the abundance of gene *ureC* among treatments as based on Tukey’s HSD test, *p* values (** *p* < 0.01, *** *p* < 0.001)

**Figure 3 microorganisms-09-00161-f003:**
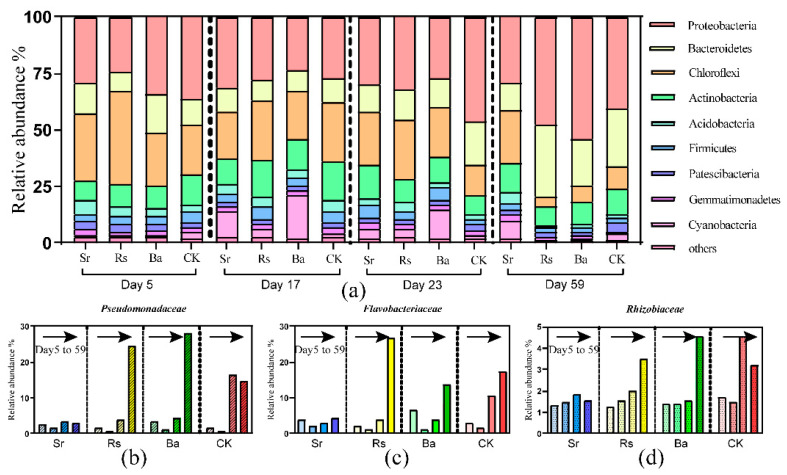
(**a**) Taxonomic comparison with relative abundance at the phylum level under the four treatments during four plant growth stages; (**b**) Change in the relative abundance of *Pseudomonadaceae* under the four treatments during four plant growth stages; (**c**) Change in the relative abundance of *Flavobacteriaceae* under the four treatments during four plant growth stages; (**d**) Change in the relative abundance of *Rhizobiaceae* under the four treatments during four plant growth stages.

**Figure 4 microorganisms-09-00161-f004:**
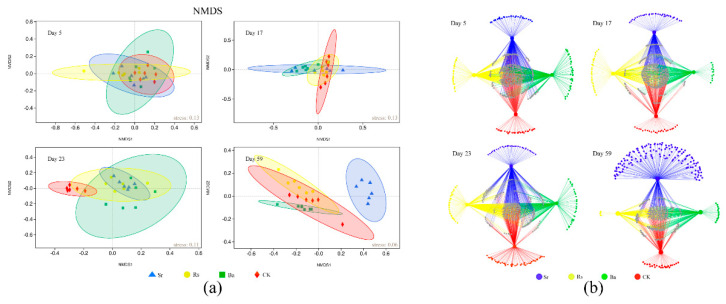
(**a**) Non-metric multidimensional scaling (NMDS) analysis of bacterial community structure under four treatments during four plant growth stages. Ellipses in the plots denote 95% confidence intervals for the centroids of different treatments. NMDS analysis is shown along two primary dimension-reduced axes. Axes were determined by non-metric multidimensional scaling and are presented in Bray-Curtis dissimilarity units. Colors indicate the treatment; (**b**) Bipartite association network, the large nodes represent different treatment groups, the small nodes represent unique OTUs belong to each treatment, the color represents the OTUs shown in the legend and the small gray nodes represent the common OTUs.

**Figure 5 microorganisms-09-00161-f005:**
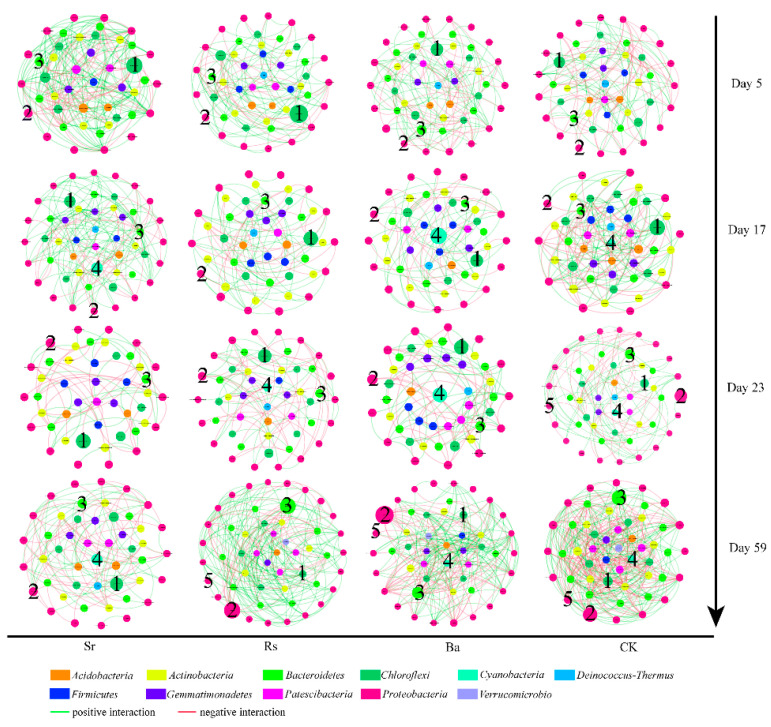
Networks at the genus level. The size of each node is proportional to the abundance of the genus it represents. Node color corresponds to phylum-level taxonomic classification. Edge color represents positive (green) and negative (red) interactions. Genera classified as (1) *Chloroflexi*; (2) *Pseudomonas*; (3) *Flavobacterium*; (4) *Cyanobacteria*; and (5) *Rhizobium*.

**Table 1 microorganisms-09-00161-t001:** Quantification PCR primers of the PGPR.

Species	Primer Sequence	Origin	PCR Product	Reference
*S. rhizophila*	TCTCAACCTGGGTACCGTAATA	rpfX	87-bp	This study
	AGATGTCCAGGCAACAGTTC			
*R. sphaeroides*	GCCTCGGCCAAGACCAACC	gyr B	250-bp	[51]
	GCTCGCCGGTGATGAAGATGGG			
*B. amyloliquefaciens*	TGGCGCCATGAGAATCCT	pgs B	66-bp	[50]
	GCAAAGCCGTTTACGAAATGA			
_	CAGACCGACGTGTGCGAAAG	*nxrA*	320-bp	[52]
	TCCACAAGGAACGGAAGGTC			
_	AAAGGYGGWATCGGYAARTCCACCAC	*nifH*	457-bp	[53]
	TTGTTSGCSGCRTACATSGCCATCAT			
_	TGGGCCTTAAAATHCAYGARGAYTGGG	*uerC*	327-bp	[54]
	GGTGGTGGCACACCATNANCATRTC			

**Table 2 microorganisms-09-00161-t002:** Data and comparison of physicochemical parameters of plants from the four treatments.

	Plants	Statistics ^§^
	Sr	Rs	Ba	CK	Sr vs. CK	Rs vs. CK	Ba vs. CK
**Chlorophyll ^#^**	42.78 ± 1.14	39.59 ± 0.87	45.66 ± 1.27	42.29 ± 0.96	ns	ns	ns
**Biomass (g) ^†^**	1.17 ± 0.07	1.35 ± 0.11	1.35 ± 0.07	1.08 ± 0.09	ns	ns	ns
**Soluble starch (mg.g^−1^) ^†^**	11.22 ± 0.32	9.89 ± 0.28	12.32 ± 0.15	10.18 ± 0.95	ns	ns	ns
**Soluble protein (mg.g^−1^) ^†^**	24.25 ± 0.30	22.59 ± 0.54	26.47 ± 0.17	21.92 ± 0.42	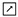 *	ns	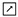 **
**Vitamin C (mg.g^−1^) ^†^**	0.43 ± 0.02	0.42 ± 0.03	0.43 ± 0.02	0.47 ± 0.02	ns	ns	ns
**TC (g.kg^−1^) ^†^**	301.51 ± 0.80	271.67 ± 3.59	300.65 ± 0.65	306.78 ± 1.55	ns	*** 	ns
**TN (g.kg^−1^) ^†^**	53.30 ± 1.59	64.11 ± 0.56	66.93 ± 3.02	47.86 ± 0.13	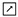 **	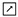 ***	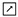 ***
**TS (g.kg^−1^) ^†^**	9.68 ± 0.10	10.40 ± 0.10	10.95 ± 0.25	10.79 ± 0.35	ns	ns	ns

^#^ Values are mean ± SE; *n* = 30. ^†^ Values are mean ± SE; *n* = 3. ^§^ HSD test. Significant levels: ns: *p* > 0.05; * *p* < 0.05; ** *p* < 0.01; *** *p* < 0.001. 
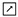
 represents increase, 

 represents decrease.

**Table 3 microorganisms-09-00161-t003:** Comparison of bulk and rhizosphere soil chemical and physical properties from the four treatments on day 59.

	Sr	Rs	Ba	CK	Statistics ^§^
	Rhizosphere	Bulk Soil	Rhizosphere	Bulk Soil	Rhizosphere	Bulk Soil	Rhizosphere	Bulk Soil	Sr Rhizosphere vs. Bulk	Rs Rhizosphere vs. Bulk	Ba Rhizosphere vs. Bulk	CK Rhizosphere vs. Bulk	Sr vs. CK rhizosphere	Rs vs. CK rhizosphere	Ba vs. CK rhizosphere
**pH ^∫^**	7.71 ± 0.13	7.50 ± 0.09	7.67 ± 0.13	7.52 ± 0.05	7.55 ± 0.10	7.30 ± 0.15	7.49 ± 0.03	7.45 ± 0.03	ns	ns	ns	ns	ns	ns	ns
**Ca** **(mg.g^−1^) ^∫^**	2923.44 ± 17.42	2775.27 ± 34.63	3020.09 ± 55.99	3031.11 ± 41.29	2697.36 ± 30.48	3138.69 ± 48.92	3159.29 ± 37.36	3218.87 ± 20.09	ns	ns	*** 	ns	* 	ns	*** 
**Mg (mg.g^−1^) ^∫^**	684.41 ± 20.36	589.71 ± 23.29	828.84 ± 38.11	909.06 ± 12.27	559.76 ± 23.44	850.69 ± 28.21	816.57 ± 29.05	855.39 ± 10.84	ns	ns	*** 	ns	** 	ns	*** 
**Fe** **(mg.g^−1^) ^∫^**	2251.47 ± 16.43	2312.01 ± 17.51	2419.31 ± 58.83	2422.01 ± 33.97	2124.92 ± 38.91	2473.94 ± 33.77	2365.15 ± 24.4	2473.95 ± 8.06	ns	ns	*** 	ns	ns	ns	*** 
**Mn (mg.g^−1^) ^∫^**	59.44 ± 0.31	60.16 ± 0.28	64.72 ± 2.75	62.36 ± 0.67	55.32 ± 1.02	62.92 ± 0.37	63.05 ± 0.96	65.15 ± 1.22	ns	ns	** 	ns	ns	ns	** 
**Al** **(mg.g^−1^) ^∫^**	1021.41 ± 10.44	968.59 ± 5.58	1032.47 ± 9.91	1028.72 ± 7.6	999.21 ± 7.97	1048.67 ± 6.38	1018.19 ± 7.39	1035.32 ± 3.6	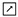 ***	ns	** 	ns	ns	ns	ns
**K (mg.g^−1^) ^∫^**	1784.97 ± 17.7	1622.75 ± 15.68	1917.26 ± 41.16	1790.56 ± 21.41	1728.42 ± 49.49	1838.31 ± 19.79	1839.89 ± 18.98	1878.54 ± 13.7	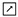 **	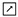 *	ns	ns	ns	ns	ns
**P (mg.g^−1^) ^∫^**	250.85 ± 3.93	201.7 ± 6.12	280.34 ± 17.47	242.94 ± 7.69	211.48 ± 10.01	188.09 ± 9.17	202.21 ± 6.57	168.02 ± 1.7	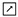 **	ns	ns	ns	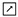 *	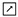 ***	ns
**NH_4_^+^ (mg.kg^−1^) ^∫^**	20.94 ± 2.63	3.65 ± 0.08	12.95 ± 1.52	3.69 ± 0.36	11.21 ± 1.22	4.62 ± 0.12	25.79 ± 1.08	4.47 ± 0.1	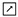 ***	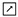 ***	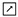 *	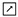 ***	ns	* 	** 
**NO_3_^−^ (mg.kg^−1^) ^∫^**	146.92 ± 3.37	90.26 ± 2.48	115.48 ± 7.99	107.74 ± 16.58	90.43 ± 6.00	146.34 ± 13.17	156.09 ± 4.99	149.84 ± 3.37	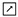 **	ns	** 	ns	ns	** 	* 
**TC** **(g.kg^−1^) ^∫^**	21.51 ± 0.82	20.42 ± 0.3	28.53 ± 2.9	22.26 ± 0.33	20.99 ± 0.53	20.8 ± 0.27	23.85 ± 0.46	19.73 ± 0.54	ns	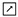 **	ns	ns	ns	ns	ns
**TN** **(g.kg^−1^) ^∫^**	1.45 ± 0.09	1.35 ± 0.04	2 ± 0.22	1.6 ± 0.07	1.43 ± 0.07	1.48 ± 0.06	1.51 ± 0.14	0.98 ± 0.15	ns	ns	ns	ns	ns	ns	ns
**TS (g.kg^−1^) ^∫^**	0.33 ± 0.02	0.52 ± 0.02	0.41 ± 0.05	0.57 ± 0.02	0.32 ± 0.02	0.44 ± 0.03	0.33 ± 0.02	0.41 ± 0.03	*** 	** 	* 	ns	ns	ns	ns

^∫^ Values are mean ± SE; *n* = 6. ^§^ HSD test. Significant levels: ns: *p* > 0.05; * *p* < 0.05; ** *p* < 0.01; *** *p* < 0.001. 
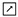
 represents increase, 

 represents decrease.

## Data Availability

The data presented in this study are openly available in the NCBI Sequence Read Archive, the accession number PRJNA622875.

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
