# Peer review of "Unraveling Mechanisms and Impact of Microbial Recruitment on Oilseed Rape (Brassica napus L.) and the Rhizosphere Mediated by Plant Growth-Promoting Rhizobacteria"

_microorganisms, 2021, doi:10.3390/microorganisms9010161_

Round 1

Reviewer 1 Report

The manuscript submitted to me for review examines microbial relationships, and changes in the rhizosphere and bulk soil niches of Brassica napus amended with three previously selected PGPR strains. There are three treatments with inoculants and one control in the pot experiment in a greenhouse. Soil samples were taken during four plant growing stages from bulk and rhizosphere soil from each treatment. It can be seen that statistical analyzes of the obtained results have been made, but it is not very clear what exactly. It may be a good idea to include a section in the Materials and Methods summarizing how this analysis is made with which type of software.

The manuscript is well organized and meets the requirements of Instruction to authors of the journal. Each of the sections is well developed and detailed.

Analyses of nutrient elements and compounds, chemical and physical properties revealed differences between treatments provoked by PGPR’s populations. Meanwhile, the growth-promoting abilities of the three strains were assessed.

Metagenomic analyses were conducted that reveals the weight of each phylum during plant growth among treatments. Finally, Non-metric multidimensional scaling and ecological networks of dynamic interactions between bacterial communities were presented.

In my opinion, the overview of the literature could be presented more in deep, revealing the most important findings not only for the applied strains but also for others that apply the approach used in the current study.

The thesis is correctly constructed, the research is consistent, and the results predispose to answer the questions. The methodology is clearly explained, and the obtained results are connected to the theory. The manuscript is well written and with sufficient explanations.

In addition, a wealth of supplementary material is presented, which makes it possible to get a complete picture of the research done.

The sentence of l. 40 needs to be reworked.

l. 45: The sentence is too general to be correct in the corresponding sense. Some of them are PGPR because they are able to promote plant growth.

l. 102: the pore size of 0.22 µm should be written correctly. Otherwise, it may cause confusion.

l.116: The sentence needs to be reworked. The verb is missing.

Section 2.8 has to be rewritten due to a lot of generalities.

Please check the results in Table 2 and especially the significance level in Rs treatment for TN.

l. 210: Here, you report results of elemental concentrations in bulk and rhizosphere soil, but maybe you mean available concentration and not total, but if not, the available one shall be presented. Please specify.

Reviewer 2 Report

Dear Colleagues!

I would like to highlight the scientific value of the manuscript, dealing with the fate of PGPR based inoculants in the rhizosphere. Please find below several observations and remarks.

Introduction:

Rows 57-58: The sentence “How the composition of rhizosphere microbiome changes after inoculation with B. amyloliquefaciens, R. sphaeroides and S. rhizophila” has no sense. Please reformulate the objective of the present study.

Material and methods:

Row 72: Please use italics when scientific names are used (Brassica campestris)

Row 106: Please indicate the purpose of further processing of the soil samples

Rows 181-187: a paragraph from the instructions for authors remained in the text. Please delete form the text.

Discussions:

Rows 520-523: a paragraph from the instructions for authors remained in the text. Please delete from the text.

Best wishes for your further work!

Reviewer 3 Report

The manuscript by Yung Li investigated the effect of microbial recruitment mechanisms induced by three PGPR strains, Stenotrophomonas rhizophila, Rhodobacter sphaeroides, and Bacillus amyloliquefaciens on ecophysiological properties of Oilseed rape rhizosphere and bulk soil. They showed that inoculation with S. rhizophila, R. sphaeroides, and B. amyloliquefaciens, significantly increased plant total N content. Both R. sphaeroides and B. amyloliquefaciens selectively enhanced the growth of Pseudomonadacea and Flavobacteriaceae, whereas S. rhizophila promoted diazotrophic rhizobacteria. Initial colonization by PGPR in the rhizosphere affected the composition of the rhizobacterial community all along the plant life cycle. Lastly, the concluded about the existence of a PGPR species-dependent driving microbial competition mechanism in the rhizosphere, which potential application to increase plant growth.

Overall, the paper contains a lot of very interesting new data but required to be carefully presented before its publication.

I/ Major Queries:

Q1: The abstract contains 145 words could be much better presented by highlighting additional finding here (still 55 words left)

Q2: The introduction section is very short (542 words) nearly the double of what is needed for the abstract.

Q3: Why the authors did not conduct study using consortium composed of either: Ba-RS; BA-Sr; Rs-Sr. or even all of them?

Q4: However, there is a concern related to the editing of the paper as it needs to be carefully checked for typos and to respect the editing guide of the journal. Several sentences are not complete which makes difficult to follow-up the contents.

Q4: the list of references with 154 ones: is too much. There is a need to reduce it for example by citing reviews…..

II/ Minor Queries (Mq): The list is not exhaustive and must be extensively checked.

Mq1: Line 69: 20h add space 20 h, the same al along the text, ml by mL etc 

Mq2: line 80: 16cm to 16 cm……

Mq3: Line 141: containing 0.4μl FastPfu Polymerase: we need to see the number of enzymatic Unit used instead of giving a volume

Mq4: Line 161: Escherichia coli has to be in italic…..the same must be applied for all bacterial strain along the text.

Mq5: Table 1. xxbp to be replaced by xx-bp and also elsewere

Mq7: Line 171: end of the sentence: 25 ul to be replaced by 25 μl.

Mq8: Lines 182-187: delete paragraph corresponding to microorganism’s instruction: starting from Material & Methods to the end.

Mq9: Lines: 190-191: Sr and Ba have to be written not in abbreviation as are cited in the first time in the result section.

Mq:10 Line 221: the last sentence is incomplete

Mq: 11: Line 390 last word: by make to be replaced by: by making…….

Mq: 12: Lines:  520 to 523: delete sentence form Authors to the end  as it corresponds to the Authors instruction.

Once corrected, It will be easier for me  to go again through this interesting  manuscript.  Maybe I should have additional constructive remarks.

Round 2

Reviewer 3 Report

The revised version is fine for me.  The authors provide convincing answers to all queries. Only few minor changes listed below are required: 

Lines 102  and 103: /g must be replaced buy g-1

Lines 152 to 156, add space when required :

''was amplified in a 20μL reaction volume containing 2.5 units FastPfu Polymerase, 4μL 5×FastPfu 153 buffer, 2.5 mM dNTPs, 0.2μL BSA (all TransGen, China), 0.5 μM forward primer and reverse primer. 154 PCR was performed using the following PCR program: 95℃ for 3min, 27 cycles of 95℃ for 30s, 55℃ 155 for 30s, 72℃ for 45s, followed by 72℃ for 10min''

Line 191:  replace 25 uL by 25μL

The paper is now suitable for publication.
